

# Effects of lower and upper body fatigue in striking response time of amateur karate athletes

Júlio Cesar Carvalho Rodrigues[1], Eduardo Macedo Penna[1,2],
Hugo Enrico Souza Machado[1,2], Jader Sant'Ana[3],
Fernando Diefenthaeler[3] and Victor S. Coswig[2,4]

[1] Faculdade de Educação Fisica, Universidade Federal do Pará, Castanhal, Pará, Brazil
[2] Programa de Pós-Graduação em Ciências do Movimento Humano, Universidade Federal do Pará, Castanhal, Pará, Brazil
[3] Laboratório de Biomecânica, Universidade Federal de Santa Catarina, Florianópolis, Santa Catarina, Brazil
[4] Instituto de Educação Física e Esportes, Universidade Federal do Ceará, Fortaleza, Ceará, Brazil

Corresponding author
Victor S. Coswig, vcoswig@ufc.br

## ABSTRACT

In combat sports, strikes or counter-strikes response time (RT) can be related to performance and sporting success. Moreover, training sessions are usually highly fatiguing, which is expected to impair basic skills, such as RT. Thus, this study aimed to investigate the effect of fatigue on punch and kick RTs of karate practitioners. Twelve individuals of both sexes from different levels (three yellow belts, three red belts, two orange belts, two green belts, one brown belt, and one black belt) were selected. Participants were aged $22 \pm 3$ years old, with a stature of $169.1 \pm 6.5$ cm, and a body mass of $65.5 \pm 10$ kg. Six visits were held with each participant. On the first 2 days, the RT of punches and kicks was measured by a validated smartphone app (TReaction). For the subsequent visits, a randomized incremental test for the upper or lower body was adopted as motor fatigue protocol, immediately followed by punches and kicks RT tests, also in random order. For induction of lower and upper body-specific muscle fatigue, the ITStriker app was used, which operates by emitting sound signals transmitted by a smartphone. One-way repeated measures ANOVA was performed, and significance was set at $p \leq 0.05$. Regarding the mean punches RT, significant effects between situations for the upper ($F_{(2,22)} = 11.5$; $\omega^2 = 0.23$; $p < 0.001$) and lower body ($F_{(2,22)} = 14.2$; $\omega^2 = 0.18$; $p < 0.001$) fatigue protocols were found. The negative effect of the lower body fatigue protocol in punches RT was evident regardless of the order of the tests (punch RT first: $\Delta = 10.5\%$; t = 4.4; $p < 0.001$; $d = 1.0$; kick RT first: $\Delta = 11.4\%$; t = 4.8; $p < 0.001$; $d = 1.1$). Regarding mean kicks RT, significant effects were found between situations for the lower ($F_{(2,22)} = 16.6$; $\omega^2 = 0.27$; $p < 0.001$) but not for the upper ($F_{(2,22)} = 2.3$; $\omega^2 = 0.02$; $p = 0.12$) body fatigue protocols. Kick RTs were negatively affected by the lower body fatigue protocol regardless of the RT order applied (punch RT first: $\Delta = 7.5\%$; t = 3.0; $p = 0.01$; $d = 0.8$; kick RT first: $\Delta = 14.3\%$; t = 5.7; $p < 0.001$; $d = 1.5$). Upper body fatigue does not impair punch or kick RTs. Thus, it is concluded that the specificity of fatigue protocols and striking order should be considered while performing RT demanding techniques in karate practice. Specifically, lower body motor fatigue may impair both kicks and punches RT, which highlights the role of lower limbs in

punches performance. Otherwise, upper body motor fatigue seems to induce impairments that are limited to the specific motor actions of this body segment.

# INTRODUCTION

Karate is a striking combat sport in which movements are performed intermittently, with high-intensity actions interspersed by short recovery/low-intensity periods. The scores are mainly achieved through punches and kicks (*Glaister, 2005*; *Chaabène et al., 2012*). In kumite matches (*i.e.*, sparring), fast reactions are required due to the high speed of the striking actions and the proximity between opponents. Therefore the notion that experienced karate athletes would have better response time (RT) than novice practitioners relies on contradictory findings (*Mori, Ohtani & Imanaka, 2002*; *Martinez-de-Quel et al., 2015*). RT is calculated from the time between the presentation of an initial stimulus and the execution of the motor gesture (*Hultsch, MacDonald & Dixon, 2002*). In the context of striking combat sports, those points can be stated from an external visual or audible stimulus to the end of a strike reaching a target (*Sant' Ana et al., 2017*), which in turn, would represent the sensorial perception of an opponent action (*i.e.*, hip movement) and its consequent motor response (*e.g.*, striking, dodging, or blocking), respectively.

Specifically, the construct of RT seems to be composed of central and peripheral variables, that refer to (i) the time between the stimulus and the onset of muscle activation (pre-motor reaction time) and the full contraction (RT) and; (ii) the time between the onset of muscle activation and the full contraction (*Liu et al., 2022*). In this sense, RT may be affected by cognitive (*e.g.*, reaction time) and motor (*e.g.*, maximal voluntary force) factors, which in turn may respond differently depending upon the status and characteristics of specific fatigue processes (*Bounty et al., 2011*). As a consequence, an inability to maintain muscle strength and impulse levels during repeated muscle contractions can be expected (*Halson, 2014*). Also, essential capacities such as aerobic and anaerobic endurance, and decision-making performance can be altered by fatigue as well (*Gierczuk et al., 2012*).

Recently, the fatigue phenomenon has been described as a psychophysiological condition in which motor and/or cognitive performances showed impairments, and can increase its respective fatigue perceptions (*Behrens et al., 2022*). Thus, it could be expected that a given task would induce a fatigue state in light of the specificity principle, that is, an activity-induced fatigue state would be dependent on the context of the task. For example, different ergometers (*i.e.*, non-motorized treadmill *vs.* cycling) can induce specific fatigue responses to a similar effort, which may be related to motor performance fatigue factors such as the volume of muscle mass involved and the contraction mode (*Kerhervé et al., 2020*; *Behrens et al., 2022*). Specifically, regarding the effects of predominant upper body tasks on lower body performance, *Dunn et al. (2021)* showed that intermittent rowing performed at maximum intensity impaired the performance of punches in highly trained

boxers. However, investigations on the effects of specific fatigue in striking RT are still emerging. For example, *Sant' Ana et al. (2017)* showed that a kick fatigue protocol could induce adverse responses in reaction time and kick's impact. However, the question of how punches and kicks RT differently respond to different motor fatigue protocols remains unclear.

Also, the available evidence is restricted to measures of RT that not necessarily can be applied to the sports-specific context (*Martinez-de-Quel et al., 2015*; *Pavelka et al., 2020*). Therefore, investigating the effects of specific fatigue induction on the RT of punches and kicks of karate fighters could contribute to a better understanding of the effect of specific karate protocols and provide more effective decision-making on physical and technical-tactical training for karate. Thus, the present study aimed to investigate the effect of specific lower body or upper body muscular fatigue on the RT of punches and kicks in karate fighters. The central hypothesis is that the induction of motor fatigue can negatively affect the RT of punches and kicks according to the specificity principle.

## MATERIALS AND METHODS

### Experimental approach to the problem

Six laboratory visits were held with each participant, separated by a minimum of 24 h between visits. Initially, on the first 2 days, the RT for punches (*Guiako zuki*) and kicks (*Mawashi-geri*) were measured and the mean value was considered as baseline. In the subsequent four sessions, the incremental fatigue tests for the lower body and upper body were applied in a randomized sequence. Immediately following the lower body and upper body fatigue induction, the punch and kick RT were measured considering the dominant limb, and the sequence was also randomized. The experimental design is described in Fig. 1.

### Subjects

This study was approved by the Institute of Health Sciences ethics committee from the Federal University of Pará (#5.539.827) and participants signed the informed consent form prior to participation. The research project was advertised at karate gyms in the city. A total of 12 individuals of both sexes from different levels (three yellow belts, three red belts, two orange belts, two green belts, one brown belt, and one black belt) were selected. Participants were aged $22 \pm 3$ years old, with a stature of $169.1 \pm 6.5$ cm, and body mass of $65.5 \pm 10$ kg. As inclusion criteria, the athlete should have practiced the sport for at least 6 months and regularly trained at least three times per week, practicing 1 h and 30 min per training session or longer. The exclusion criteria were musculoskeletal joint problems, muscle and/or joint discomfort during the progressive fatigue tests, occurrence of muscle injury in the previous 3 months, or the athlete's absence during the meetings to perform the tests. Participants were asked to avoid strenuous exercise for, at least, 48 h before experimental days.
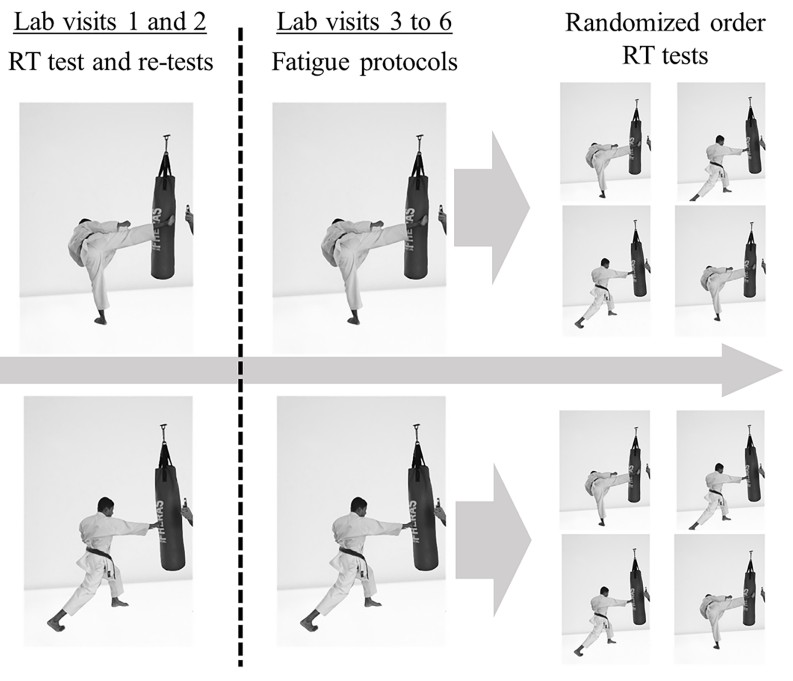

**Figure 1 Experimental design.** RT, Response time 

## Procedures

### Response time

To measure RT, the TReaction® app (ETS4ME, São José, SC, Brazil) was used, which is calculated from a visual stimulus (flash) emitted by a smartphone. The visual stimulus was captured by the individual who performs the strike until it reaches the target (punch bag), and the sound generated by the strike's impact on the target is captured by the smartphone microphone, ending the RT. The app has been validated to measure the response time of combat sports athletes (ICC = 0.998) (*Coswig et al., 2019*). In this sense, five direct punching techniques were executed with the opposite hand to the stance (*guiako zuki*). In addition, five dominant-side roundhouse kicks were landed in a region marked by the evaluator (*mawashi-geri*). The interval from the emission of the stimuli to the execution of the five punch attempts varied between 10 to 15 s. The distance between the athlete and the punch bag was set following the protocol described in the app validation study (*Coswig et al., 2019*). Briefly, for the punch test distance, the athletes positioned themselves laterally to the punch bag and abduct the shoulder with a clenched fist until touch the target. For the kick test, a frontal position was adopted with the leg lifted at a 90° angle until the foot sole touch the punch bag. Both distances were maintained until the tests end.

### Incremental tests

For induction of lower and upper body-specific muscle fatigue, the ITStriker app was used, which operates by emitting sound signals transmitted by a smartphone or tablet (*Sant' Ana, Sakugawa & Diefenthaeler, 2021*) using validated incremental kick and punch protocols (*Sant' Ana et al., 2019*). For the lower body test, the participant was asked to stand at a distance where the kicking motion would reach the kicking bag. The athletes

**Table 1 Delineation of the punches and kicks incremental tests used to induce muscle fatigue in upper and lower body, respectively.**

| Stage | Duration (s) | Accumulated duration (s) | Pace (beeps per stage) | Number of kicks of the stage | Number of punches of the stage |
|---|---|---|---|---|---|
| 1 | 100 | 100 | 6 | 6 | 12 |
| 2 | 84.0 | 180 | 10 | 10 | 20 |
| 3 | 77.1 | 260 | 14 | 14 | 28 |
| 4 | 73.3 | 330 | 18 | 18 | 36 |
| 5 | 70.9 | 405 | 22 | 22 | 44 |
| 6 | 69.2 | 470 | 26 | 26 | 52 |
| 7 | 68.0 | 540 | 30 | 30 | 60 |
| 8 | 67.1 | 605 | 34 | 34 | 68 |
| 9 | 66.3 | 675 | 38 | 38 | 76 |
| 10 | 65.7 | 740 | 42 | 42 | 84 |
| 11 | 65.2 | 805 | 46 | 46 | 92 |
| 12 | 64.8 | 870 | 50 | 50 | 100 |

were also asked to stay in a stepping movement, simulating a fight. For fatigue induction, the athlete had to respond with a kicking motor action at the opponent's chest height whenever the sound signal from the ITStriker app was triggered. The protocol was progressive, and the time interval between beeps decreased progressively, as described in Table 1. The protocol was finished when, the athlete could no longer follow the frequency of the beeps (pace), and/or did not reach the height pre-established by the researcher, and/ or voluntarily stopped the test (volitional exhaustion). This procedure was the same as reported in previous studies (*Sant' Ana et al., 2017*, *2019*; *Sant' Ana, Sakugawa & Diefenthaeler, 2021*). For the upper body specific test, the same app was used, which has also applied to boxers (*de Oliveira et al., 2022*). The upper body fatigue protocol functioned exactly as mentioned above, using the validated incremental punching protocol (*Sant' Ana et al., 2019*). The participants positioned themselves in front of a punching bag, using karate specific gloves, at a distance favorable to their armspan, which was self-selected. Once the distance was chosen by the participant, it was maintained until the end of the test following the evaluator's feedback. For each sound signal, the participant started a sequence of two punches: a front punch (*kizami zuki*) and then a back punch (*guiaku zuki*).

### Rate of perceived exertion

For measuring the subjective rate of perceived exertion (RPE) the Borg CR-10 scale was used, which ranges from level 0 (no exertion) to 10 (maximal exertion) (*Kurilla, 2011*). At the end of each cycle, the participant was asked to answer, "How hard was your training session?" on a 0 to 10 scale (*Foster et al., 2001*).

### Statistical analysis

After verifying the normality of the data by using the Shapiro-Wilk test, the data are presented by mean, as a measure of centrality, and standard deviation, as a measure of dispersion. Test and retest response times reliability was tested by intraclass correlation

**Table 2 Performance during the incremental punches and kicks tests and retests.**

| | Kicks | | Punches | | ANOVA | | |
| --- | --- | --- | --- | --- | --- | --- | --- |
| | Test | Retest | Test | Retest | F | p-value | $\omega^2$ |
| Pace$_{MAX}$ (n) | 29.9 ± 1.9 | 29.4 ± 2 | 33.1 ± 4.3 | 31.9 ± 5.6 | 2.92 | 0.05 | 0.09 |
| Total number of strikes (n) | 126.8 ± 15.3 | 122.7 ± 15.9 | 290.7 ± 101.9[#] | 310.8 ± 75.7[#] | 31.61 | <0.001 | 0.64 |
| Total test duration (min) | 8.7 ± 0.6 | 8.5 ± 0.6 | 9.4 ± 1.5 | 9.6 ± 1.3 | 2.69 | 0.09 | 0.08 |
| Rating of perceived exertion (a.u.) | 9.1 ± 1.2 | 9.0 ± 1.5 | 9.5 ± 1.0 | 9.5 ± 0.8 | 1.09 | 0.33 | 0.004 |

Notes:
[#] Statistically different from Kicks test and retest at $p < 0.001$ (Cohen's d ranged from 2.5 to 2.9).
Pace$_{MAX}$, amount of motor action relative to the end of the last stage reached; n, number of strikes; a.u., arbitrary units.

coefficient (ICC). The type of ICC used was the "two-way mixed effects, absolute agreement, multiple measurements" as it was recommended for test-retest and Intrarater Reliability Studies (*Koo & Li, 2016*). The mean between the test and retest was taken as the baseline value and used for comparisons between time points. Comparisons between time points were checked by one-way ANOVA for repeated measures with the Bonferroni *post hoc* test when appropriate and, when sphericity was not reached, the Greenhouse-Geisser correction was considered. The magnitude of the differences was verified from the Cohen's d effect size and were then scaled as trivial (0.50–1.0), or large effects (>1.0) (*Rhea et al., 2003*). All tests were carried out in IBM SPSS Statistics for Windows, version 20.0 (IBM Corp., Armonk, N.Y., USA). Statistical significance was established when $p < 0.05$.

## RESULTS

From the 12 participants included, nine were male and five were experienced athletes (above green belt), while seven were considered as moderate experience (from yellow to orange belt). Descriptive information stratified by sex and experience is presented in Fig. S1. No significant correlations were found between RT and performance in the incremental tests.

Regarding the performance in the incremental test, values are described in Table 2. Despite the expected significant differences found regarding total number of strikes (two punches and one kick were performed at each beep), no differences were observed in Pace$_{MAX}$, total test duration or RPE at the last test stage.

Considering baseline values, no significant differences were observed between the test and retest for punches mean RT ($p = 0.26$; $d = 0.33$) or kicks mean RT ($p = 0.28$; $d = 0.32$). Regarding ICC and its respective confidence intervals, a value of 0.75 (0.19–0.92) was found for punches mean RT, while an ICC of 0.75 (0.17–0.92) was identified for the mean kick RT.

Figure 2 shows the mean punches RT in response to fatigue protocols. The repeated measures ANOVA showed statistically significant effects between situations for the upper ($F_{(2,22)} = 11.5$; $\omega^2 = 0.23$; $p < 0.001$) and lower body ($F_{(2,22)} = 14.2$; $\omega^2 = 0.18$; $p < 0.001$) fatigue protocols. The *post hoc* tests suggested that punches RT is negatively affected when is tested immediately after the upper body fatigue protocol ($\Delta = 11.8\%$; t = 4.8; $p < 0.001$; $d = 1.4$). However, the negative effect of the lower body fatigue protocol in punches RT was

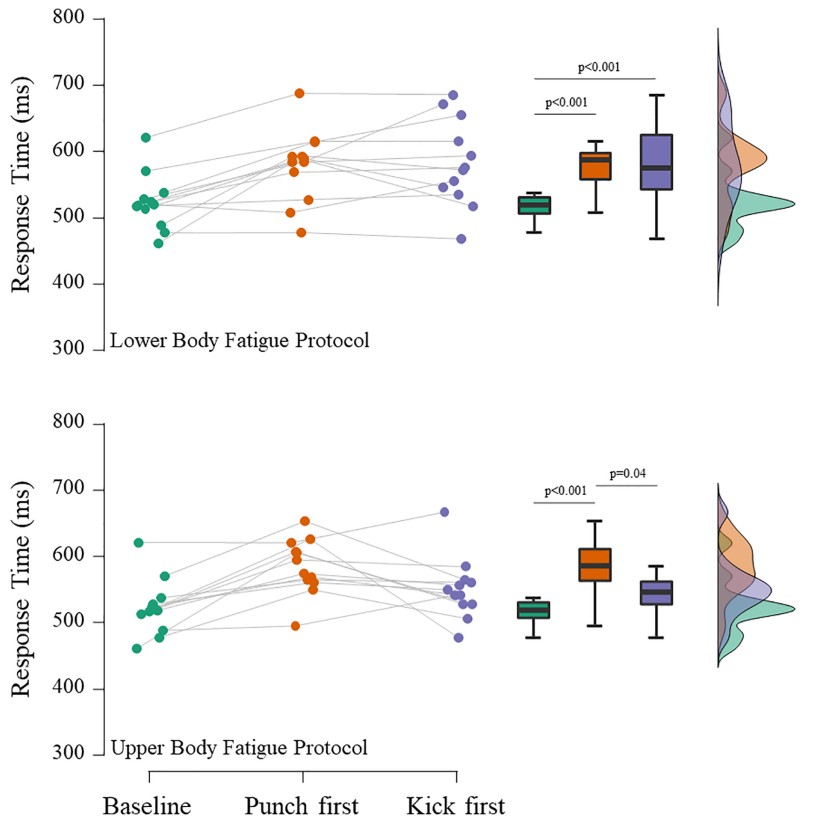

**Figure 2 Mean punch (*guiaku zuki*) Response Time in different experimental situations.**

evident regardless of the order of the tests (punch RT first: $\Delta = 10.5\%$; $t = 4.4$; $p < 0.001$; $d = 1.0$; kick RT first: $\Delta = 11.4\%$; $t = 4.8$; $p < 0.001$; $d = 1.1$).

Figure 3 shows the mean kicks RT in response to fatigue protocols. The repeated measures ANOVA showed statistically significant effects between situations for the lower ($F_{(2,22)} = 16.6$; $\omega^2 = 0.27$; $p < 0.001$) but not for the upper ($F_{(2,22)} = 2.3$; $\omega^2 = 0.02$; $p = 0.12$) body fatigue protocols. The *post hoc* tests suggested that kicks RT were negatively affected by the lower body fatigue protocol regardless of the RT order applied (punch RT first: $\Delta = 7.5\%$; $t = 3.0$; $p = 0.01$; $d = 0.8$; kick RT first: $\Delta = 14.3\%$; $t = 5.7$; $p < 0.001$; $d = 1.5$). Also, when kicks RT are tested right after the fatigue protocol, the results seem to be worsened ($\Delta = 6.3\%$; $t = 2.7$; $p = 0.03$; $d = 0.7$). On the other hand, upper body fatigue does not impair punches or kicks RT.

## DISCUSSION

The main hypothesis of this investigation was that the induction of motor fatigue could negatively affect the RT of punches and kicks according to the specificity principle. The findings indicated that the hypothesis was confirmed. RT of kicks was only affected by the specific incremental kick test, while, alternatively, the RT of punches was more susceptible to fatigue effects regardless of the motor pattern generating fatigue.

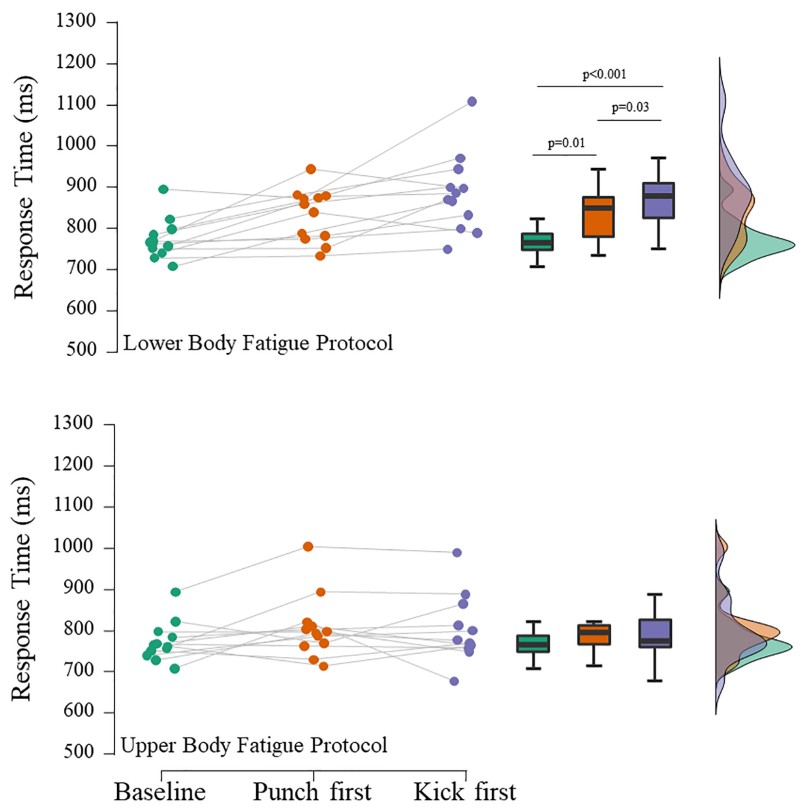

**Figure 3 Mean kick (*mawashi geri*) Response Time in different experimental situations.**

Beyond the correspondent effects of upper and lower body fatigue in punches and kicks performance, respectively, the crossed effect of lower body fatigue in punches RT was also expected. The role of lower body muscles in punching performance was already described and supports the notion that "leg drived" actions may be related to higher force production in boxers (*Dinu & Louis, 2020*). Indeed, lower extremity jumping training was shown to be an interesting alternative to improve punching performance, including reaction time, peak ground force reaction of the rear leg, and punch velocity (*Chottidao et al., 2022*), while squats and squat jumps were effective to provide potentiation effects on the rear-hand straight punch performance (*Yi et al., 2022*). Also, a recent review suggested that lower body, but not upper body strength, was related to higher punching impact (*Beattie & Ruddock, 2022*). Together, this evidence highlights the relevance of lower body actions on punching performance, which may rely on the expectation that impairments provoked by specific site motor fatigue can influence the opposite segment, as shown by Dunn (*Dunn et al., 2021*). Thus, considering the role of the lower body on the mentioned variables (*e.g.*, reaction time, peak ground force reaction of the rear leg, punch velocity, and punch impact) and its relation with RT, our findings suggest that fatigue impairments in this segment can be harmful to punching RT.

Our results are not without precedent findings. Fatigue induced by kick execution was shown to negatively affect RT ($145 \pm 51$ ms to $223 \pm 133$ ms; $p < 0.05$) and impact ($43 \pm 27$ g to $13 \pm 10$ g; $p < 0.01$) of kicks in taekwondo athletes after a specific fatigue protocol

using roundhouse kick (*Sant' Ana et al., 2017*). In the present study, the intensities reached by the participants were similar to the fatigue induction protocol used in abovementioned study (individual maximum intensity strike rate). Hence, the participants reached the maximum pace and exhaustion in the maximum incremental tests immediately before the RT tests. In contrast, when the effort was performed at an intensity relative to the anaerobic threshold, no impairment effect was observed in any of the variables related to RT and neither on the performance of the vertical jump (*Sant' Ana, Sakugawa & Diefenthaeler, 2021*). Therefore, it appears that the increase in the kick RT (*i.e.*, kick execution time) and the maintenance of the movement's technical quality and performance, might be influenced by the intensity demand imposed by the intermittence relation of the high and low-intensity actions that the athlete is submitted to. Thus, the fatigue process (or tolerance to it), the quality of the motor pattern, and the RT result from the modulation or demand required for the intervention or combat to which the subject is submitted (*Hermann et al., 2008*; *Gierczuk et al., 2012*; *Sant' Ana, Sakugawa & Diefenthaeler, 2021*).

In the present study, the upper body fatigue protocol negatively affected the RT of punches. Similar results were observed with MMA athletes who showed worsening RT after performing the Wingate test to induce upper body fatigue (*Pavelka et al., 2020*). Additionally, the RT of punches was also impaired when the athletes were subjected to lower body fatigue. Similar findings were observed in a study by *Dunn et al. (2021)* that examined the induction of lower body and trunk region fatigue by rowing test that resulted in reduced impact and punching strength of boxers. Collectively, these findings suggest that induction of fatigue in the lower body is more systemic than upper body protocols, which would be expected since lower body efforts move more muscle mass, thus producing higher values of $O_2$ consumption (*Andreato et al., 2022*).

In striking combat sports, the decisive actions are supplied by the anaerobic energy system, particularly the ATP-CP system, noting that these actions require high force production in short time intervals (*Andersen & Aagaard, 2006*). In addition to being dependent on an increased number of muscle fibers recruitment, especially fast-twitch (*Maffiuletti et al., 2016*). Thereby, the fact that an increased RT after the lower body incremental test was found, might be related to the contractile function of the involved muscles, thus directly affecting the ability to respond in the same way as they were before the fatigue protocol (*MacIntosh & Rassier, 2002*). Previous studies have showed this ability to respond by the fatigue index, speed reduction, and drop in the impact of blows in a specific test and its relation between shorter blows time and vertical jumps (*Sant' Ana et al., 2014*), as well as has been seen on boxing fighters that presented reductions on force and power outputs after a rowing test.

In general, kick RT scores were impaired independently on the incremental protocol. It is suggested that more complex motor gestures may require more cortical areas to be involved in processing tasks, thus considering that kicking is a more complex task than punch, it may explain why kicks were more easily affected (*Snyder & Cinelli, 2022*). Moreover, the differences observed after the upper body and lower body incremental protocols may be related to mechanisms associated with the fact that both protocols

present different motor gestures. Another explanation may rely on the task duration, since RT, but not reaction time, seems to be mainly affected by ongoing processes throughout the movement and not by the movement initiation itself (*Donovan et al., 2006*). Notwithstanding, kicking has ~40% higher RT than punching, which would suggest a higher susceptibility of kicks to be affected by other factors, such as feedback and fatigue. However, to establish a conclusion on how the fatigue process occurs physiologically and its effects on RT, additional studies are needed due to the complexity of the different mechanisms associated with fatigue (*Enoka & Duchateau, 2008*) and in particular with karate athletes, experience level (*Quinzi et al., 2013*).

As a limitation, karate practitioners of non professional competitive level were part of this study. Consequently, the findings cannot be extrapolated to professional competitive levels, especially for aspects related to technical consistency. A further limitation was the absence of specific measures that could better describe the systemic or neuromuscular fatigue conditions presented by the athletes after the incremental efforts, such as measures of oxygen consumption, muscle activity, vertical jump tests, and local RPE, for example. However, the inclusion of different orders of RT tests after fatigue induction helped to improve meaningful inferences that should be considered in future work. Finally, although essential for this experimental design, specific gestures for fatigue induction (punching or kicking) need to reflect the pattern of training and fights (punching and kicking), which may limit inferences related to fatigue derived from more usual daily practices. Furthermore, the outcomes presented as group average, with different experience and expertise levels, may have contributed to the results.

## CONCLUSIONS

The findings presented here can be useful for coaches, trainers, and practitioners while planning and performing karate training sessions. In general, martial arts training is composed of highly physically demanding warm-ups and technique repetitions. However, our results suggest that, when punches and kicks RT are relevant elements of the training session, some combinations should be avoided.

Thus, considering that kick performance may be impaired by previous lower body fatigue, coaches should avoid extenuating and high-volume stimuli or guarantee enough recovery before the training main activity, or evaluations. Regarding punch performance, it seems more sensitive to non-specific fatiguing actions, such as those performed mainly by the lower body segment, and for this reason, coaches should dedicate attention to avoiding excessive demanding efforts with lower and upper body actions. However, if the preceding actions are performed predominantly by the upper body, adequate recovery time may be sufficient to reestablish punches RT.

In summary, after taking the limitations of heterogeneity of our sample into account, our findings are relevant for the striking combat sports environment. With this, coaches and trainers can adjust the training to improve the technical and tactical performance, avoiding excessive and exhausting use of the lower body in karate training to obtain a better performance of the RT of kicks and punches.

Finally, it was concluded that the specificity of fatigue protocols should be considered while performing RT demanding techniques in karate practice. Therefore, lower body fatigue negatively affects the RT performance of kicks when performed in the same order. Furthermore, it is worth highlighting that regarding upper body RT, the incremental test for the lower body alters the RT independently of the evaluation order, that is, whether punches or kicks RT comes first. Nonetheless, the incremental test of the upper body has just affected the RT of punches when it comes first than kicks.

## ACKNOWLEDGEMENTS

The authors would like to thank to the athletes for their time and dedication to this project.

### Funding

This work was supported by Proeg/UFPa– Pró-reitoria de ensino de graduação and Propesp/UFPa – Pró-Reitoria de Pesquisa e Pós-Graduação for student funding. Support was also received from CNPq and Capes-Brazil Programa de Bolsas Universitárias de Santa Catarina (UNIEDU) provided a scholarship for Jader Sant' Ana. The funders had no role in study design, data collection and analysis, decision to publish, or preparation of the manuscript.

### Grant Disclosures

The following grant information was disclosed by the authors:
Proeg/UFPa–Pró-reitoria de ensino de graduação and Propesp/UFPa–Pró-Reitoria de Pesquisa e Pós-Graduação.
CNPq and Capes-Brazil.
Programa de Bolsas Universitárias de Santa Catarina (UNIEDU).

### Competing Interests

Victor S. Coswig is an Academic Editor for PeerJ.

### Author Contributions

- Júlio Cesar Carvalho Rodrigues conceived and designed the experiments, performed the experiments, prepared figures and/or tables, authored or reviewed drafts of the article, and approved the final draft.
- Eduardo Macedo Penna conceived and designed the experiments, prepared figures and/or tables, authored or reviewed drafts of the article, and approved the final draft.
- Hugo Enrico Souza Machado performed the experiments, prepared figures and/or tables, authored or reviewed drafts of the article, and approved the final draft.
- Jader Sant' Ana conceived and designed the experiments, analyzed the data, authored or reviewed drafts of the article, and approved the final draft.
- Fernando Diefenthaeler conceived and designed the experiments, authored or reviewed drafts of the article, and approved the final draft.

- Victor S. Coswig conceived and designed the experiments, performed the experiments, analyzed the data, prepared figures and/or tables, authored or reviewed drafts of the article, and approved the final draft.

## Human Ethics

The following information was supplied relating to ethical approvals (*i.e.*, approving body and any reference numbers):

This study was approved by the Institute of Health Sciences ethics committee from the Federal University of Pará (#5.539.827).

## Data Availability

The raw measurements are available in the Supplemental Files.

## Supplemental Information

Supplemental information for this article can be found online at http://dx.doi.org/10.7717/peerj.14764#supplemental-information.

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
