# Peer review of "Effects of lower and upper body fatigue in striking response time of amateur karate athletes"

_PeerJ, doi:10.7717/peerj.14764_

## Round 0.1 · original submission · Minor Revisions

Dr Rodrigues and co-authors. The Reviewers (and I) have identified a number of issues in your manuscript which requires your attention. Please ensure you address each of the points in a Response document and make the changes to your manuscript with "track changes" within the next 21 days please.

·

Basic reporting

Dear authors,

Your submitted paper “Effects of lower and upper body fatigue in striking response time of amateur karate athletes” is beneficial for a deep view of this issue. However, for improvement, consider the notes below.

Rewriting two paragraphs in the introduction; More specifically focused studies related to the research conducted should be added; Add outcomes of the individuals because of different participants´ experiences; In the discussion is missed explicit comment on the stated hypothesis; more limitations should be added.

Experimental design

The methods were described well but there could be a problem with the different experience levels of the participants. The results should also be adapted to this, where the results of individual participants should also be displayed

Validity of the findings

The obtained results are beneficial for the field of combat sports. However, consider adding specific literature to the discussion.

Additional comments

Dear authors,

Your submitted paper “Effects of lower and upper body fatigue in striking response time of amateur karate athletes” is beneficial for a deep view of this issue. However, for improvement, consider the notes below.

Abstract
Row 26 - The study included participants from different levels of karate (from yellow to black belt). I think that labeling the respondents as amateurs is not appropriate.

Introduction

Row 47 - there is a mention of anticipatory skills related to RT and the next paragraph (row 52-59) compose only of points, where is not any notice of anticipation which, as is known, is also affected by orientation in space and experience of participants.

Row 67 - 70 - is a stated sentence that emphasizes the need for a better understanding of the effect of the fatigue protocol and next are stated examples from individual studies. I recommend rewriting paragraphs 3 (row 60-67) and 4 (row 67-76) in such a way that they make gradual sense and result in a clear need, what exercise protocols need to be carried out and not just stacking a selection of studies one behind the other and concluding everything with the sentence that "it is not a direct measure of sports-specific skills, such as the RT of kicks and punches". I understand what you want to express that there were conducted several studies that included the non-specific fatigue protocol (Jump, Wingate test, etc.) but not in connection with basic techniques of combat sport (stroke and kick). However, this part of the paper is needed more precisely so that it is related to subsequent analysis of the research and discussion

Row 82 - Hypothesis "The central hypothesis is that the induction of fatigue can negatively affect the RT of punches and kicks according to the specificity principle." There is mention of the specificity principle. From the text of the introduction, it is not clear that this principle can have an effect on RT, it is related to rewriting paragraphs 3 and 4 and clearly preparing the basis for such a hypothesis

Methods

Row 101 - There can be a problem with a clear conclusion related to the composition of the respondents' group. The combination of gender and participants' level in the group where there are only 12 participants. Who will the conclusions of the studies be aimed at?

Row 105 - I recommend state mention in the exclusion criteria previous, no muscle injury in the previous three months

Row 142 - add a reference to the prediction equation used

Results
Row 166 – the abbreviation (n) should be described under Table 1
Row 168 – the lower and upper confidence intervals should be stated after CI

The part of the statistical analysis is stated that the ANOVA was used but in the interpretation of the results is not obvious its use, only the values of paired tests are given

Table 3 – Puch; correction to Punch (left title of the table in part A)

Because there was a group of participants with different experiences (from yellow up to the black belt), I recommend adding the outcomes, table, or graph, where will be outcomes of the individual participants. I checked the raw data and there are obvious differences between the participants

Discussion

There should be answered the hypothesis that was stated in part of the introduction

Row 231-242 - the essence of this research is focused on specific fatigue in combat sports, however, there is overall little work with published research in combat sports, such as Quinzi 2013, who mentions skill-dependent activation strategy and neuromuscular adaptation related to kicking and their kinematic indicators.

Limitation
I recommend adding a mention that was not found the efficiency of the kicks and punches (dynamic forces and techniký průběh provedení), next, that the outcomes are presented for the group with different experiences (from the yellow up to black belt)

·

Basic reporting

The basic reporting of this manuscript meets some of the standards required, although some elements of statistical results reporting are missing. These have been highlighted and requested in the additional comments section below.

Experimental design

This design is original and clearly described within the text. There are some minor additions requested in the additional comments that might serve to make it easier for future readers.
The research question is defined, but with one ambiguous phrase. It has been requested that this phrase is better clarified in the additional comments to ensure the aims of the study and the hypothesis are clear.

Validity of the findings

All raw data have been provided, with ethical guidelines adhered to. Additional statistical results have been requested in the additional comments to demonstrate robustness, improve replicability and use of the results in future research.

Additional comments

To the authors,
Thank you for submitting this manuscript investigating the effects of fatigue protocols on response time in amateur karate athletes. This is an interesting paper which may provide some useful information for coaches working with competitive karate athletes. There are, however, some improvements to be made to the manuscript before it can be accepted for publication. Please review my comments on the introduction and discussion as suggestions, with my comments on the methods and results as requirements (unless there is a reasonable reason why these cannot be met). I look forward to reading the next version of this manuscript.
Kind regards,
Christopher Kirk

General comments
Please replace , for . in all data reporting to avoid confusion for the international reading audience.
Consider using a different abbreviation for ‘response time’, as RT is widely used for ‘reaction time’ and may cause confusion for readers.

Introduction
Lines 48-51: Consider splitting this sentence into two and providing a brief description of what the ‘external visual or audible stimulus’ would be in this sport.
Line 57: please exchange power for impulse in keeping with more appropriate terms.
Lines 65-66: consider being specific about the mode of these lower body exercises (cycle/rowing/running ergometer) to allow these results to be interpreted in context
Lines 71-75: What were findings of these studies? Did RT and jump height increase or decrease?
Lines 82-83: Please clearly state what is meant be ‘according to the specificity principle’ here. Does this mean you expect fatigue to occur during the execution of the same skill that has just been used in the fatigue protocol?

Material and Methods
Lines 101-102: please state the n recruited for each belt and sex. And please change height for stature in keeping with the use of more appropriate terms
Lines 111-143: photographic figures of these set ups with the participant in position would be useful here. Please consider adding these if possible.
Line 116: please also report the previously found reliability of this app.
Lines 137-139: Please change ‘wingspan’ to ‘armspan’ in keeping with more appropriate terms. Please also state how once the participant chose their distance this was made consistent between each strike. How did you ensure the distance from the bag did not change between each strike?
Lines 141-142: What is the purpose of estimating V̇O2max here? What relationship does this have to the measurement of RT in this study? Please justify this, and if you decide to keep it in the paper, please provide a reference for the source of this regression equation. If there is no justification relating to the specific hypothesis then please remove.
Line 146: please replace ‘effort’ with ‘exertion’ as these are two different concepts.
Line 154: Please state which type of ICC was calculated in keeping with recommendations for correct use of this type of testing (https://www.sciencedirect.com/science/article/pii/S1556370716000158?casa_token=OUBqxnhPFp4AAAAA:o1rAMCrydulvPGNwOQX2vnWdxcfQjzFCy_ZAi4neQKllTNlh4txQ0ppdbKVz0Kpcahd3U_LSxQ)
Please calculate and report effect sizes for the paired tests (Cohen’s d) and ANOVA (omega squared) to allow for comparison between studies, determination of power for future studies and for inclusion in future meta-analyses.

Results
Please report degrees of freedom for all t tests, and all F and degrees of freedom for all ANOVA results to allow results checking. And please see previous comment regarding effect sizes.
Figures are well laid out, but please consider adding a jitter element to each bar on Figures 2 and 3 to show the full range and density of the sample if possible. This would enable a more contextual reading of the data to occur.
Please check the p value for the V̇O2max in Table 1 – is it possible to have a p value greater than 1.0?

Discussion
Lines 201-205: Please consider rewording this sentence as it is difficult to understand what point is being made. This might be because of the phrase ‘the intensity of maximum strike frequency was applied’. Please rephrase.
As it currently stands, the discussion is relatively superficial and repeats some of the information from the introduction. Please provide a more detailed discussion of the potential reasons for these results from a physiological perspective (where possible). This would provide a better lead into the practical training recommendations mentioned in the conclusion.

---

## Round 0.2 · accepted · Accept

Thank you for addressing all of the changes identified by the Reviewers. I am pleased to recommend your manuscript be accepted for publication. Thank you for your support of PeerJ and we look forward to receiving future manuscripts.